# One Health Approach to the Computational Design of a Lipoprotein-Based Multi-Epitope Vaccine Against Human and Livestock Tuberculosis

**DOI:** 10.3390/ijms26041587

**Published:** 2025-02-13

**Authors:** Robert Adamu Shey, Gordon Takop Nchanji, Tangan Yanick Aqua Stong, Ntang Emmaculate Yaah, Cabirou Mounchili Shintouo, Bernis Neneyoh Yengo, Derrick Neba Nebangwa, Mary Teke Efeti, Joan Amban Chick, Abey Blessings Ayuk, Ketura Yaje Gwei, Arnaud Azonpi Lemoge, Luc Vanhamme, Stephen Mbigha Ghogomu, Jacob Souopgui

**Affiliations:** 1Department of Biochemistry and Molecular Biology, Faculty of Science, University of Buea, Buea P.O. Box 63, Cameroon; tangan.yanick@ubuea.cm (T.Y.A.S.); mayaahemma@gmail.com (N.E.Y.); nebaderrick@gmail.com (D.N.N.); tekeefetimary@gmail.com (M.T.E.); abeyayuk8@gmail.com (A.B.A.); keturayaje2@gmail.com (K.Y.G.); stephen.ghogomu@ubuea.cm (S.M.G.); 2Tropical Disease Interventions, Diagnostics, Vaccines and Therapeutics (TroDDIVaT) Initiative, Buea P.O. Box 1022, Cameroon; nchanji.gordon@ubuea.cm; 3Department of Microbiology and Parasitology, Faculty of Science, University of Buea, Buea P.O. Box 63, Cameroon; 4Department of Microbiology and Immunology, College of Medicine, Drexel University, 2900 W Queen Ln, Philadelphia, PA 19129, USA; shintouocabi@gmail.com (C.M.S.); bernisyengo@gmail.com (B.N.Y.); 5Frailty in Ageing Research Group, Vrije Universiteit Brussel, Laarbeeklaan 103, B-1090 Brussels, Belgium; 6Department of Gerontology, Faculty of Medicine and Pharmacy, Vrije Universiteit Brussel, Laarbeeklaan 103, B-1090 Brussels, Belgium; 7Department of Computer and Information Sciences, College of Science and Technology, Covenant University, PMB 1023, Ota 112233, Ogun State, Nigeria; joan.chickpgs@stu.cu.edu; 8Ngonpong Therapeutics, 3640 Concord Pike #1145, Wilmington, DE 19803, USA; aazonpil@gmail.com; 9Department of Molecular Biology, Institute of Biology and Molecular Medicine, IBMM, Gosselies, Université Libre de Bruxelles, Rue des Professeurs Jeener et Brachet 12, B-6041 Charleroi, Belgium; luc.vanhamme@ulb.be (L.V.); jacob.souopgui@ulb.be (J.S.)

**Keywords:** tuberculosis, immunoinformatics, lipoproteins, TB-MEVA−1, multi-epitope vaccine candidate

## Abstract

Tuberculosis (TB) remains a major cause of ill health and one of the leading causes of death worldwide, with about 1.25 million deaths estimated in 2023. Control measures have focused principally on early diagnosis, the treatment of active TB, and vaccination. However, the widespread emergence of anti-tuberculosis drug resistance remains the major public health threat to progress made in global TB care and control. Moreover, the Bacillus Calmette–Guérin (BCG) vaccine, the only licensed vaccine against TB in children, has been in use for over a century, and there have been considerable debates concerning its effectiveness in TB control. A multi-epitope vaccine against TB would be an invaluable tool to attain the Global Plan to End TB 2023–2030 target. A rational approach that combines several B-cell and T-cell epitopes from key lipoproteins was adopted to design a novel multi-epitope vaccine candidate. In addition, interactions with TLR4 were implemented to assess its ability to elicit an innate immune response. The conservation of the selected proteins suggests the possibility of cross-protection in line with the One Health approach to disease control. The vaccine candidate was predicted to be both antigenic and immunogenic, and immune simulation analyses demonstrated its ability to elicit both humoral and cellular immune responses. Protein–protein docking and normal-mode analyses of the vaccine candidate with TLR4 predicted efficient binding and stable interaction. This study provides a promising One Health approach for the design of multi-epitope vaccines against human and livestock tuberculosis. Overall, the designed vaccine candidate demonstrated immunogenicity and safety features that warrant further experimental validation in vitro and in vivo.

## 1. Introduction

Tuberculosis (TB), caused by species of the *Mycobacterium tuberculosis* complex (MTBC), remains a major cause of ill health and one of the leading causes of death worldwide [1]. While TB in humans is caused principally by *Mycobacterium tuberculosis* sensu stricto, other highly related subspecies of the *Mycobacterium tuberculosis* complex, such as the bovine *Mycobacterium bovis*, can also cause human TB [2]. It has been estimated that 1.4% of all human TB cases in the world and 2.8% of all cases in the African population are attributed to *M. bovis* [3]. Since bovine TB can spread from cattle to humans and from humans to cattle, there is an urgent need to adopt the One Health approach to control the disease. *M. bovis* has one of the broadest host ranges of all known pathogens, affecting many groups of mammals [4]. Susceptible species include cattle, humans, non-human primates, goats, cats, dogs, pigs, buffalo, badgers, possums, deer, and bison [5]. TB is prevalent in all countries, affecting all age groups but mostly adults in their productive years, leading to a significant social and economic impact [6]. The WHO estimated that 10.8 million people were affected by TB worldwide in 2023, the majority of whom were men (about 56%). In addition, a total of 1.25 million deaths were reported (including 161,000 people with HIV). Furthermore, 1.3 million cases were children [7], representing a significant number, since healthcare professionals frequently ignore tuberculosis in children and adolescents, and it can be challenging to identify and treat [8].

Tuberculosis is preventable and curable, and control measures have focused principally on early diagnosis, the treatment of active TB (to stop its spread), and the prevention of active disease in exposed individuals or those with a known latent infection [9]. It is estimated that up to 66 million lives were saved through TB diagnosis and treatment between 2000 and 2020 [7]. The currently approved TB diagnostic tests, encompassing smear microscopy, microbiological cultures, and molecular methods, all rely on sputum but suffer from poor diagnostic accuracy [10]. In addition, treatment is focused on using a combination of antibiotics for periods of between six and 12 months [11]. However, the long periods needed for treatment raise concerns about compliance and increase the emergence and persistence of antimicrobial resistance [12]. In fact, anti-TB drug resistance remains the major public health threat to the progress made in TB care and control worldwide [12,13].

To further compound the challenges faced by control measures, the Bacillus Calmette–Guérin (BCG) vaccine, the only licensed vaccine against TB in children, has been used for over a century [14], and there have been considerable debates concerning its effectiveness in TB control [15]. It has been reported that, while infant BCG shows moderate efficacy in the prevention of severe extrapulmonary TB in young children [16], its efficacy in preventing tuberculosis in adults and adolescents in several clinical trials has shown large variation, and the vaccine has been largely ineffective in controlling the global epidemic [17].

The achievement of the WHO’s strategic target of ending the TB epidemic by 2035 urgently requires novel, safe, and effective vaccines [18]. Multi-epitope vaccines (MEVs) represent an innovative strategy based on immunoinformatics methods, aiming to design more robust vaccines for infectious diseases, with several advantages [19]. Compared to classical vaccine approaches, which have been largely empirical, reverse vaccinology has several advantages, including the selection of epitopes that can induce strong cellular and humoral immune responses simultaneously, the elimination of unwanted antigen components that could trigger either pathological immune reactions or adverse effects, and the possibility to polarize the immune response through the introduction of specific agonist for pathogen recognition receptors [20]. Several MEVs have been designed for various diseases, including TB [21,22,23]. The development of a robust multi-epitope vaccine depends primarily on the selection of appropriate candidate antigens and their immunodominant epitopes [24]. Bacterial lipoproteins exhibit several functions, including nutrient uptake, signal transduction, virulence, adhesion, conjugation, and sporulation, and participate in antibiotic resistance and transport [25,26]. The development of immune responses against *Mycobacterium tuberculosis* lipoproteins could block vital physiological processes in the bacteria. This work focused on the deployment of immunoinformatics tools to design a novel MEV candidate for MTB based on selected lipoproteins reported to be implicated in pathogenesis and virulence.

## 2. Results

### 2.1. Protein Retrieval and Preliminary Characterization

The sequences of 14 lipoproteins (P9WIF5, O53692, Q79FB3, P9WIB5, P9WJE1, P9WK61, P9WK65, P9WNF3, P9WK45, P9WG29, O53859, I6Y3P1, A5TZX4, and P9WGT7) previously reported to be involved in pathogenesis or virulence constituted the initial dataset [27,28]. Among the obtained sequences were proteins that had already been investigated as vaccine candidates, including ESAT-6-like protein (O53692) [29] and PE_PGRS33 (P9WIF5) [30]. The sequences of all selected lipoproteins were downloaded from the UniProtKB database and used for preliminary characterization to down-select the antigens to be employed for epitope prediction and chimeric antigen design. Nine of the selected proteins were predicted to have signal peptides by the SignalP 6.0 server, while the TOPCONS server predicted signal peptides in 12 of the selected proteins. For TM domain prediction, only one protein (O53859) was predicted by consensus using TOPCONS to have one TM region. The DeepTMHMM server 1.0, on the other hand, predicted no TM domains in all selected proteins. The DeepLocPro-1.0 server predicted six proteins to be localized in the extracellular space. Seven proteins were predicted to be localized in the cytoplasmic membrane, while one (P9WK65) was predicted to be localized on both the cell wall and cytoplasmic membrane. Meanwhile, on the TBpred server, most of the proteins were predicted to be integral membrane proteins (6). Moreover, some proteins were predicted to be attached to the membrane by a lipid anchor (4). In addition, others were predicted to be localized in the cytoplasm (2) or secreted (2) (Appendix A). Following these analyses, seven antigens (P9WK61, P9WK65, P9WNF3, P9WK45, I6Y3P1, A5TZX4, and P9WGT7) were predicted to have a signal peptide and be localized in the extracellular space by the DeepLocPro-1.0 server. They were then selected for epitope prediction analyses (Table 1). The *Mycobacterium tuberculosis* 50S ribosomal L7/L12 protein (RL7_MYCTU), P9WHE3, reported to be a TLR4 agonist [31], was downloaded from the UniProtKB database to be deployed as a built-in adjuvant.

### 2.2. Protein Conservation in Other Mycobacterium Species

All of the selected proteins showed a high level of conservation in related species of the *Mycobacterium* genus. Percentage identities ranging from 30.9 to 100% were observed following sequence alignment analyses against the UniProtKB database and the BLASTp search on NCBI. Two proteins, I6Y3P1 and P9WNF3, had no homologs in *Mycobacterium leprae*. The conservation of the proteins against their homologs in *M. decipiens, M. leprae, M. lacus, M. gordonae, M. asiaticum, M. bovis, M. riyadhense,* and *M. pseudokansasii* was performed. The highest level of conservation was observed for *M. bovis*, with six of the seven proteins showing 100% conservation and the last protein showing 99.8% conservation (Table 2).

### 2.3. Cytotoxic T-Lymphocyte (CTL) Epitope Prediction

High-scoring cytotoxic T-lymphocyte (9-mer) epitopes were predicted from the seven selected antigens (in Table 1) on the NetCTL 1.2 server using the set threshold of 1.0. A total of sixteen CD8+ epitope sequences were selected for incorporation into the chimeric antigen based on their antigenicity scores or because they overlapped with other predicted HTL or LBL epitopes (Table 3).

### 2.4. Helper T-Lymphocyte (HTL) Epitope Prediction

HTL epitopes were chosen from among high-affinity MHC-II epitopes predicted on the NetMHCII 2.3 web server for the HLA-DR, HLA-DQ, and HLA-DP human alleles (based on IC50 scores). To design the chimeric vaccine candidate, a total of eight high-affinity HTL epitopes were selected. Some of the predicted linear B-cell epitopes and HTL epitopes were found to have overlapping regions (Table 3).

### 2.5. Linear B-Lymphocyte (LBL) Epitope Prediction

Two servers were used to predict linear B-lymphocyte epitopes (with different numbers of amino acid residues), and epitopes concurrently predicted by both servers were selected to design the multi-epitope chimeric vaccine candidate. One of the predicted linear B-lymphocyte epitopes had a region of overlap with a predicted HTL epitope, while another also had a region of overlap with one of the predicted CTL epitopes. A total of nine linear B-epitopes were selected to be incorporated into the chimera (Table 3).

### 2.6. Immunoglobulin Class Prediction for LBL Epitopes

All nine selected LBL epitopes predicted to be antigenic and chosen for incorporation in the designed chimeric antigen were subjected to antibody class prediction using the AbCPE server. All nine epitopes were predicted to bind to IgG. None of the selected epitopes was predicted to bind to IgE or IgM, while two (from proteins A5TZX4 and P9WK65) were predicted to induce IgA. The prediction probabilities ranged from 66 to 100%, with four LBLs demonstrating a 100% probability of inducing antibody production (Table 4).

### 2.7. Design of the Chimeric Multi-Epitope Vaccine Candidate

Predicted epitopes containing overlapping regions were joined to form contiguous sequences (Table 1). Consequently, the multi-epitope chimeric vaccine candidate was designed from 11 CTL epitopes, 7 HTL epitopes, and 9 linear B-lymphocyte epitopes in total. The chimeric vaccine candidate was generated by incorporating AAY, GPGPG, and KK linkers to merge the CD8+ T-epitopes, CD4+ T-epitopes, and linear B-cell epitopes, respectively. In addition, the TLR4 (PDB ID: 4G8A) agonist, *M. tuberculosis* 50S ribosomal protein L7/L12 (P9WHE3), was selected as a built-in adjuvant. The built-in adjuvant was integrated at the N-terminus of the designed chimeric vaccine candidate by the EAAAK linker to enhance the elicited immune responses. Moreover, the Pan DR Epitope (PADRE) sequence (AKFVAAWTLKAAA), an effective stimulator of CD4+ responses, was incorporated after the built-in adjuvant sequence. Furthermore, the TAT sequence (GRKKRRQRRRPQ) was added after the PADRE sequence using a GGGS linker. The 10xHis tag was incorporated at the carboxyl terminal of the designed chimeric antigen to facilitate antigen purification (by affinity chromatography) and identification (by Western blot). Summarily, the chimeric vaccine candidate (designated TB-MEVA−1) comprised 683 amino acid residues, which included 27 *M. tuberculosis* predicted epitopes, the incorporated immune potentiators (TLR4 agonist, TAT, and PADRE), and diverse linkers (Figure 1).

### 2.8. Physicochemical Properties and Solubility Analysis

The designed chimeric antigen, composed of 683 amino acids, was predicted to have a molecular weight (MW) of 69.7 kDa and a theoretical pI of 9.36, indicating that the protein was basic. In addition, physicochemical property analyses revealed the protein to be rich in Ala (15.2%) and Gly (12.7%). The predicted half-life of the designed protein was 30 h in vitro in mammalian reticulocytes, >20 h in yeast, and >10 h in vivo in *E. coli*. The instability index (II) was predicted to be 27.03. This score classifies the protein as stable since it is below the threshold score of 40.00. The predicted grand average of hydropathicity (GRAVY) was −0.295, indicating the potential of the designed protein to interact with water molecules. The predicted aliphatic index was 76.31, suggesting that the protein exhibited thermostability. The predicted solubility score was 0.4811 (slightly below the 0.5 threshold) on the NetSolP 1.0 server and 0.4218 on the DeepSoluE server, suggesting that the designed chimera may only be marginally soluble following expression in a bacterial host (Table 5).

### 2.9. Secondary Structure and Intrinsic Disorder Prediction

The designed vaccine candidate protein was predicted to consist of 27% alpha helices, 13% beta strands, and 58% coils (Figure 2A). Considering solvent accessibility, 59% of the amino acid residues were predicted to be exposed, while 16% and 23% were predicted to be medium-exposed and buried, respectively. Thirty-one percent (217 amino acid residues) were predicted to be located in intrinsically unstructured regions (Figure 2B). Figure 2C displays a graphic representation of the secondary structure according to the color of the designed multi-epitope vaccine candidate.

### 2.10. Prediction of Antigenicity, Allergenicity, and Toxigenicity

The antigenicity score of the designed vaccine candidate was predicted to be 1.1314 on the VaxiJen 2.0 server (with a bacteria model threshold set at the 0.4 default threshold). The antigenicity score on the ANTIGENpro server was 0.9218. In addition, the Vaxi-DL server predicted the designed chimera to be a vaccine candidate with a 96.66% probability. The results obtained suggest that the designed vaccine candidate is antigenic. In addition, the AllerTOP v.2 and AllergenFP servers both predicted that the designed protein was non-allergenic. Meanwhile, the ToxinPred and ToxDL servers predicted that TB-MEVA−1 did not contain any toxic peptides, with a score of 0.0001 predicted on the ToxDL server (Table 6).

### 2.11. IFN-γ-Inducing Epitope Prediction

Of the 122 potential IFN-γ-inducing epitopes (15-mer) predicted for the built-in adjuvant, 62 scored above the epitope prediction threshold. Meanwhile, of the 486 potential IFN-γ-inducing epitopes predicted for the main vaccine sequence, 86 had scores above the set cut-off. The large number of predicted IFN-γ-inducing epitopes showed a correlation with the level of IFN-γ induction following immune simulation analyses on the C-ImmSim server with the designed vaccine candidate (Figure 3).

### 2.12. Prediction of Mouse MHC II Epitopes

From the 669 9-mer epitopes that could be derived from the antigen for each allele, 168 peptides that could bind six mouse MHC II alleles were predicted in the TB-MEVA−1 antigen. The H-2-IAb allele was observed to have the highest number of epitopes (46.7%), while the H-2-IEk allele had no binding epitopes (Table 7).

### 2.13. Prediction, Refinement, and Verification of the Modeled 3D Structure

AlphaFold2 was used to predict the functional 3D structure of TB-MEVA−1 using the ColabFold interface (Figure 4A). The predicted 3D structure was refined twice, first on the ModRefiner server and then by the GalaxyRefine server, which yielded five models. From the five predicted models, based on the obtained results, “model 4” was selected as the final 3D model for further characterization (Figure 4B). The selected model had the following scores: GDT-HA (0.8679), RMSD (0.682), and MolProbity (1.072). Additionally, the clash score was 1.6, the poor rotamers score was 0.0, and the Ramachandran plot score was 97.1%. The Ramachandran plot score predicted on the GalaxyRefine server was similar to the 97.06% of residues predicted to be located in favored regions by the PDBsum server. Furthermore, the 3D structure was predicted to have 2.1% of residues located in allowed regions, and only 0.2% of residues were predicted to be found in disallowed regions (Figure 4C). After 3D structure refinement, the quality was verified using the ProSA-web and ERRAT servers. The ProSA-web server predicted refined a Z-score of −4.92 (Figure 4D), while the ERRAT server predicted an overall quality factor of 94.444. For native proteins of comparable size, the predicted ProSA-web score fell outside of the usual score range.

### 2.14. Discontinuous B-Cell Epitope Prediction

Three hundred and seventy-one amino acid residues (53.1%) in the designed chimeric antigen were predicted to be found in 25 conformational B-cell epitopes using the ElliPro server. The epitope prediction scores ranged from 0.517 to 0.983, with the number of residues in the predicted discontinuous epitopes ranging from 3 to 68 amino acid residues (Appendix A).

### 2.15. Protein–Protein Interaction Between the TLR4 Receptor and the Designed Vaccine Candidate

A suitable immune response depends on a coordinated interaction between epitopes and antigen-specific immune receptors. Different studies have reported the involvement of TLR-4 in the generation of protective immune responses against *M. tuberculosis* [32,33]. Molecular docking investigated the binding interaction between the refined and validated 3D structure of the chimeric vaccine candidate and TLR4 after a protein–protein interaction pocket was predicted in the designed chimera (Figure 5A). Thirty models were generated, displayed, and ranked according to the cluster size generated by the ClusPro 2.0 server, and the docking complex with the largest cluster size was selected. The TB-MEVA−1-TLR4 complex, with the largest cluster size, exhibited a center binding score of −1388.4, with the lowest energy identified at −1497.4 (Figure 5B). For the further characterization of vaccine–receptor interactions, the selected model of the vaccine–TLR4 complex was analyzed for its binding affinity using PrODIGY and the interactions at the interface using PDBsum. The relative binding free energy (ΔG) of the vaccine candidate–TLR4 complex was predicted to be −15.3 Kcal/mol. The predicted K_d_ was 1.7 × 10^−11^. Consistently, the number of contacts formed at the interface (IC) per property was determined (ICs charged–charged: 16, ICs charged–polar: 17, ICs charged–apolar: 35, ICs polar–polar: 6, ICs apolar–apolar: 30). For the vaccine–TLR4 complex, the PDBsum server predicted that 22 hydrogen bonds, one salt bridge, and 214 non-bonded contacts were formed between 38 residues from TLR4 and 32 residues from the chimeric vaccine candidate (Figure 5C). Above all, the vaccine candidate exhibited favorable interactions with TLR4.

### 2.16. Normal-Mode Analyses

The molecular stability and functional movements of the designed vaccine candidate–TLR4 complex were characterized through normal-mode analysis (NMA) on the iMODS server (Figure 6). The deformability graph displayed peak points representing the main chain residues’ deformed regions in the vaccine candidate–TLR4 complex. The locations with hinges are regions with high deformability (Figure 6A). The association between the NMA mobility and TB-MEVA−1-TLR4 complex is demonstrated by the B-factor plot, indicating the average RMSD values of the docked vaccine candidate–TLR4 complex (Figure 6B). The B-factor values indicate the uncertainty of each atom. The calculated eigenvalue of the vaccine candidate–TLR4 complex was 2.000137 × 10^−7^, reflecting the motion stiffness related to each normal mode (Figure 6C). The eigenvalue is an estimate of the energy required to deform the structure. Every normal mode of the complex is denoted by an individual variance (purple) and cumulative variance (green) in the variance bar. The eigenvalue and variance showed a negative correlation (Figure 6D). In addition, the interacting motions between the vaccine candidate and TLR4 in the docked complex were demonstrated by a covariance matrix. The interconnected movements among different residue pairs were indicated by correlated (red), uncorrelated (white), and anti-correlated (blue) atomic movements in the docked vaccine candidate–TLR4 complex (Figure 6E). The elastic network map, which denoted atom pairs linked by springs in the vaccine candidate–TLR4 complex, was also generated. Each dot in the graph denotes a spring connecting the corresponding pair of atoms. The stiffness of each dot is indicated by its color; darker grays denote stiffer parts, whereas the lighter dots denote more flexible sections (Figure 6F). All data from the normal-mode analyses further indicate the favorable interaction and stability in the vaccine candidate–TLR4 complex.

### 2.17. Immune Simulation Analysis of the Designed Vaccine Candidate

The immune simulation predictions demonstrated the expansion of the elicited secondary responses. Theoretically, the observed trend reflects the operational development of immunological responses (Figure 7A). Elevated IgM levels characterized the immune response generated to the prime dose. In addition, significant increases in the B-cell population, together with corresponding increases in the IgM, total IgG + IgM, IgG1, and IgG2 antibody levels, were observed for the two vaccination booster doses (Figure 7A,B). The observed pattern suggests the development of immune memory, as depicted in the development of a sustained population of memory B cells (Figure 7C). Similarly, the CD4+ and CD8+ populations were also predicted to elicit more potent responses to the antigen following inoculation with the booster doses. These responses led to sustained increases in the active T-cell populations and the development of immunological memory (Figure 7D–F). Furthermore, sustained increases in the dendritic cell, macrophage, and NK cell populations were observed during the immunization period (Figure 7G–I). In summary, the designed vaccine candidate elicited both cellular and humoral immune responses, which have been reported to be critical in protection against tuberculosis.

### 2.18. Gene Sequence Codon Optimization and In Silico Restriction Cloning

Codon optimization for the gene sequence coding for the designed multi-epitope vaccine candidate was performed using the Java Codon Adaptation Tool (JCat) server (accessed on 14 September 2024) to optimize the protein expression in bacteria (*E. coli*). The sequence, which contained 2079 total nucleotides, had a codon adaptation index (CAI) of 1.0 and mean GC content of 50.7%. These results suggest the possibility of suitable expression in the selected host. In general, GC content between 30% and 70% is recommended for optimal protein expression [34]. Finally, the codon-optimized sequence was inserted in the NdeI and XhoI sites of the bacterial vector pET30a (+), using the SnapGene software v8.0.2 (Figure 8).

## 3. Discussion

Vaccines remain one of the most valuable and reliable tools in achieving the current End TB Strategy target of a 90% reduction in TB deaths and an 80% reduction in TB incidence by 2030 [35]. Considering the monetized value of health gains, modeling studies have estimated that the introduction of an adolescent/adult vaccine could generate USD 283 to 474 billion in economic benefits by 2050 [36]. However, at present, the only TB vaccine, BCG, has been in use for over 100 years, and several reports have indicated that BCG has only moderate effectiveness in preventing severe, extrapulmonary forms of TB in young children [16] and shows large variation in its efficacy in preventing TB in adolescents and adults [17]. This has resulted in BCG being largely ineffective in controlling the global TB epidemic [16]. In the search for potential alternatives to BCG, several vaccine candidates (both single antigens and antigen cocktails) have been evaluated in clinical trials [37]. These vaccines (which can be categorized into inactivated and attenuated whole-organism live vaccines, as well as viral vector and protein/adjuvant vaccines) include ID93/GLA-SE (consisting of Rv2608, Rv1813, ESXV, and ESXW), M72 (consisting of Mtb39A and Mtb32A), GamTBvac (a fusion between Ag85a and ESAT6-CFP10), AEC/BC02 (produced from the Ag85b protein and ESAT6-CFP10), and others [38]. However, none of these has yet been licensed by the WHO. Multi-epitope vaccine candidates offer numerous advantages, including the generation of a targeted immune response and the elimination or mitigation of potential unwanted (allergenic or tolerogenic) responses [39]. Several multi-epitope vaccine candidates have been designed for different infectious diseases, including onchocerciasis [40], schistosomiasis [20], and malaria [41,42].

In addition, chimeric antigens have been designed for tuberculosis, with some of them evaluated in animal models [21,38]. The design of TB-MEVA−1 exhibits a unique approach to targeting protein antigens exposed on the surface of the bacteria and plays roles in pathogenicity and virulence. Extracellularly or cell surface-localized proteins, due to their increased accessibility to the immune system, have been reported to be good vaccine candidates [43]. Since the selected lipoprotein antigens used for TB-MEVA−1’s design are implicated in virulence and pathogenicity, blocking their activity could help in both reducing the infectivity of the pathogen and also reducing the disease severity in persons who are already infected. Previously, the *Shigella* virulence antigen VirG (also known as IcsA) has been characterized as a vaccine candidate and provoked vigorous immune responses that included antibodies capable of blocking bacterial adhesion and invasion and provided high levels of protection against *S. flexneri 2a* or *S. sonnei* in mouse models [44]. In addition, BauA and OmpA, which are proteins involved in the pathogenicity of *A. baumannii,* have been targeted as vaccine candidates and showed significant decreases in the bacterial loads in the organs of vaccinated mice [45].

Importantly, the selected antigens used in designing the chimeric vaccine candidate also exhibited the potential for cross-protection, with high levels of conservation observed in homologous proteins in *Mycobacterium* species that affect humans and livestock. The current TB vaccine has been reported to promote immunity against other non-tuberculosis mycobacteria, such as *Mycobacterium leprae*, *Mycobacterium ulcerans*, *Mycobacterium avium*, *Mycobacterium intracellulare*, and *Mycobacterium abscessus* [46]. In addition, recent studies have reported that BCG can also be used to protect livestock against *M. bovis*, with a 74% reduction in bovine TB transmission and a substantial reduction in lesions in vaccinated animals [47]. The high level of conservation observed in *M. bovis* (Table 2), the causative agent for bovine TB, which also affects humans [48], implies that the designed vaccine candidate can be a useful tool in the context of the current One Health approach to preventing human and livestock tuberculosis [49].

The roles of both B and T cells have been reported in protective immunity to tuberculosis, with the involvement of different immune mechanisms [50,51,52]. The selected linear B-epitopes were predicted to potentially elicit both IgG and IgA responses, which have been reported to be important in immune-protective mechanisms against TB [53,54]. Besides adaptive immune responses, innate immune responses are also vital in protective immune responses against TB, with the important role of TLRs also reported [55]. Studies in mice have suggested the vital protective role that TLR4 plays a role in host defense against lung infection by *M. tuberculosis*, with splenocytes from infected TLR4 mutant mice demonstrating a reduced capacity to produce the protective type 1 cytokine IFN-γ upon antigen-specific stimulation [56]. Given the role of TLR4 in generating protective immune responses, a TLR4 agonist (*M. tuberculosis* 50S ribosomal protein L7/L12) was placed at the amino terminal of the designed chimeric to serve as an adjuvant. This adjuvant has been reported to potentiate immune responses through DC maturation and pro-inflammatory cytokine production. Therefore, DCs activated by the vaccine candidate could activate naïve T cells, effectively polarize CD4+ and CD8+ T cells to secrete IFN-gamma, and elicit T-cell-mediated cytotoxicity [31]. These additional immune-stimulating effects of the adjuvant could be a valuable addition to the immune responses elicited by the selected epitopes.

The predicted B-cell and T-cell epitopes in TB-MEVA−1 were fused using specialized linkers to design a multi-epitope chimeric vaccine candidate. Linker sequences are a key component of epitope-based vaccine design and have been used to boost immune responses by minimizing the formation of neo-epitopes, thereby eliminating junctional immunogenicity and improving epitope processing and presentation [57,58]. In addition to linkers between the different epitopes (AAY, GPGPG, KK, and GGGS), the EAAAK linker was added between the built-in adjuvant sequence and the fused epitopes. The EAAAK linker has been reported to augment the expression levels of bioactivity bifunctional proteins [59]. Moreover, the TAT sequence, which is a cell-penetrating peptide (CPP), and the Pan DR epitope PADRE (a synthetic 13-mer peptide capable of activating CD4+ T cells [60]) were added to the design to further potentiate the immune response elicited by TB-MEVA−1. Cell-penetrating peptides have been successfully used in the delivery of a large variety of cargo, from small particles to proteins, peptides, and nucleic acids [61]. They have been reported to be important in enhancing peptide vaccine accumulation and persistence in lymph nodes, enhancing immunogenicity [62,63]. On the other hand, CD4+ responses have also been reported to be critical for the control of *M. tuberculosis* infection [64]. Immunoinformatics analyses of the generated TB-MEVA−1 revealed the presence of large numbers of IFN-γ epitopes, which have been reported by multiple studies to play an essential role in host defense against infection with intracellular pathogens, including *M. tuberculosis* [65]. Further analyses of the designed chimeric antigen using different servers revealed that it was not only antigenic but also non-allergenic and non-toxigenic. These characteristics are important since efficacy and safety are key components of the vaccine research and development pipeline [66].

The designed chimeric antigen had a predicted molecular weight of 69.7 kDa. The predicted solubility score suggests that the protein may only be slightly soluble upon expression in *E. coli*. *E. coli* is a common host for recombinant protein expression, and the soluble expression and purification of proteins is a vital step in many biochemical and functional investigations [67]. The predicted theoretical pI of 9.36 indicates that the designed protein is basic. The predicted instability index of 27.03 and the estimated half-life upon expression in different host organisms suggest that the vaccine candidate will be stable upon expression. Moreover, the high aliphatic index predicted for the designed antigen suggests the thermostability of the protein [68], which indicates its suitability for use in low-resource settings.

Information on the secondary and tertiary structural features of the target protein is essential in vaccine design [69]. Secondary structure analyses predicted the vaccine candidate to consist predominantly of coils (58%). In addition, 31% of the amino acid residues were predicted to be located in intrinsically disordered regions (IDRs). IDRs and alpha-helical coiled coils are reported to consist of important forms of “structural antigens” with important biomedical applications, including vaccine delivery systems and drug development [70]. In addition, IDRs have been reported to be major components of several vaccine candidates, including the circumsporozoite protein (CSP) used in RTS,S against malaria [71]. The Ramachandran plot following 3D structure and refinement revealed that most of the residues were found in the favored and allowed regions (99.0%), with very few residues in the outlier region (Figure 4); this indicates that the quality of the overall model is satisfactory.

Since a TLR4 agonist was included in the designed chimera, protein–protein docking analysis and normal mode analyses were used to investigate the potential interaction between TLR4 and TB-MEVA−1. The obtained findings reveal the presence of a large pocket in TB-MEVA−1 and favorable interactions with TLR4, with the presence of different bond types (Figure 5). Since the docking and NMA investigations predicted a stable interaction, the chimeric antigen is expected to interact with the TLR4 on professional antigen-presenting cells like dendritic cells to stimulate a protective immune response. This possibility was also observed in the immune simulation analyses, which predicted an overall increase in the elicited immune responses following injection with primer and booster doses. The immune simulation analyses also demonstrated the development of memory B cells and T cells, which were sustained for several months. These memory cell pools are vital to the rapid clearance of the pathogen upon the exposure of the host to *M. tuberculosis* [50,72].

A major limitation of this work is that all of the observations reported here were generated from in silico computational studies. The validation of these data through both in vitro and in vivo investigations remains a strong necessity. The immediate next step of this project will involve the expression of TB-MEVA−1 in a suitable host for serological analyses using samples from individuals with an infection or exposed individuals to assess antibody recognition, since the vital role of antibodies has been reported in the action of the current BCG vaccine [73]. Further investigations could include immunogenicity and protective studies in mouse models, which will provide valuable information on the potential efficacy of the designed vaccine candidate in clinical trials. With the recent development of humanized mouse models (some of which have already been tested in tuberculosis vaccine studies [74]), it is expected that the results from studies generated in these mice would be more comparable to human studies. Regarding TB-MEVA−1, its potential to elicit an immune response in preclinical studies is demonstrated by its ability to bind to mouse alleles (Table 4). The results generated from the in vitro and in vivo studies of TB-MEVA−1 will provide the requisite pilot data to decide whether the vaccine candidate can be further evaluated in clinical trials.

## 4. Materials and Methods

The methodology used in this study comprised the following parts: (1) lipoprotein selection and preliminary analyses; (2) T-cell and B-cell epitope prediction from selected proteins; (3) antigenicity prediction for selected epitopes; (4) design of a multi-epitope chimeric antigen using appropriate linkers; (5) preliminary characterization of the designed chimera; (6) 3D structure prediction, refinement, and validation of the designed vaccine candidate; (7) molecular docking with TLR-4 and normal-mode analysis; (8) immune simulation prediction; and (9) codon optimization and in silico cloning for bacterial expression. Figure 9 summarizes the methodology adopted in this study in a flowchart.

### 4.1. Protein Sequence Retrieval and Preliminary Analyses

The sequences of 14 *Mycobacterium* lipoproteins (accession numbers: P9WIF5, O53692, Q79FB3, P9WIB5, P9WJE1, P9WK61, P9WK65, P9WNF3, P9WK45, P9WG29, O53859, I6Y3P1, A5TZX4, and P9WGT7) previously reported to be involved in pathogenicity and virulence [27,28] were retrieved from the UniProtKB database (https://www.uniprot.org/) in the Fasta format into a .txt file. Preliminary characterization of the sequences included analyses for (1) localization (using the DeepLocPro 1.0 server (https://services.healthtech.dtu.dk/services/DeepLocPro-1.0/, accessed on 10 September 2024) and the TBpred server (https://webs.iiitd.edu.in/raghava/tbpred, accessed on 10 September 2024)); (2) signal peptide presence (using SignalP 6.0 (https://services.healthtech.dtu.dk/services/SignalP-6.0/, accessed on 10 September 2024) and TOPCONS (https://topcons.cbr.su.se/, accessed on 10 September 202)); and (3) transmembrane domain presence (using DeepTMHMM 1.0 (https://services.healthtech.dtu.dk/services/DeepTMHMM-1.0/, accessed on 10 September 2024) and TOPCONS (https://topcons.cbr.su.se/, accessed on 10 September 202)). All proteins predicted to be localized in the extracellular space were selected for further analyses.

### 4.2. Protein Conservation Analyses

A BLAST search was performed against the UniProtKB database (https://www.uniprot.org/) (accessed on 10 September 2024) to investigate the conservation level of each of the selected proteins among other species of the *Mycobacterium* genus. This is because a high degree of conservation across the genus suggests the possibility of cross-protection [75]. UniProtKB is the principal database used to store and interconnect information from large and distinct sources. It is the most complete catalog containing protein sequences and functional annotations [76]. The degree of conservation between the selected lipoproteins and their related homologs in *M. decipiens* (human), *M. leprae* (human), *M. lacus* (human), *M. gordonae* (human), *M. asiaticum* (primates), *M. bovis* (human, cattle and other livestock), *M. riyadhense* (humans), and *M*. *pseudokansasii* (humans) was investigated.

### 4.3. Linear B-Lymphocyte (LBL) Epitope Prediction

The sequences of the selected lipoproteins were exposed to linear B-lymphocyte epitope prediction using two servers: firstly, the BepiPred-3.0 webserver (https://services.healthtech.dtu.dk/service.php?BepiPred-3.0) (accessed on 11 September 2024) and then the LBtope server (https://webs.iiitd.edu.in/raghava/lbtope/index.php) (accessed on 11 September 2024). The BepiPred server is a sequence-based tool that uses numerical representations from protein language model (LM) embeddings to vastly improve the prediction accuracy for linear and conformational B-cell epitope prediction on several independent test sets [77]. Using the server’s default threshold of 0.1512, linear B-epitopes of different lengths for the selected lipoproteins were predicted. LBtope, on the other hand, is an epitope prediction model that uses a larger dataset of validated B-cell epitopes and non-epitopes (12,063 epitopes and 20,589 non-epitopes obtained from the IEDB database) for epitope prediction. The LBtope server, which is based on SVM, is reported to have predictive accuracy that can reach up to ~81% [78]. The server assigns scores (ranging from 0 to 100%) to each of the predicted epitopes, with a higher score indicating a higher probability of being an epitope.

### 4.4. T-Cell Epitope Prediction

The NetMHCII 2.3 server (https://services.healthtech.dtu.dk/services/NetMHCII-2.3/) (accessed on 11 September 2024), whose functioning is based on artificial neural networks (ANNs), was used to predict 15-mer helper T-lymphocyte (HTL) epitopes for human alleles (HLA-DR, HLA-DQ, and HLA-DP) using the default server parameters for strong binders (SB) and weak binders (WB) [79]. Epitopes for the HLA-DQ, HLA-DP, and HLA-DR alleles were predicted for the selected proteins, and strongly binding, highly promiscuous epitopes (having the ability to bind to several alleles) were selected to design the chimera antigen.

Cytotoxic T-lymphocyte (CTL) epitopes in the selected lipoproteins were predicted on the NetCTL 1.2 server (https://services.healthtech.dtu.dk/service.php?NetCTL-1.2) (accessed on 11 September 2024). The NetCTL 1.2 server works on an algorithm that implements epitope prediction by integrating class I MHC binding, proteasomal cleavage, and the transporter associated with antigen processing (TAP) transport efficiency and predicts CTL epitopes for 12 MHC class I supertypes [80]. For this study, the epitopes restricted to the A2, A3, and B7 supertypes were predicted to attain approximate phenotypic frequency coverage of 90% in the vaccinated population [58]. Although the default threshold for CTL epitope prediction is 0.75, a threshold of 1.0 was used (to improve the epitope specificity). Predicted epitopes were sorted by their combined scores.

### 4.5. Epitope Antigenicity Prediction

To design a multi-epitope chimeric vaccine candidate capable of stimulating a robust protective immune response, the VaxiJen v2.0 server (http://www.ddg-pharmfac.net/vaxijen/VaxiJen/VaxiJen.html) (accessed on 11 September 2024) was used to assess the antigenicity of all predicted epitopes. This server uses an alignment-independent approach based on the physicochemical characteristics of peptides and proteins to classify antigens. The server algorithm functions based on the autocross-covariance (ACC) transformation of input peptide sequences into uniform vectors of principal amino acid features to predict the antigenicity of proteins from bacteria, fungi, parasites, viruses, and tumors [81]. The default threshold for the bacteria model (0.4) was used to evaluate epitope antigenicity, and epitopes with scores above 0.5 were selected to design the chimeric vaccine candidate. Figure 9 provides a flowchart showing the steps involved in this study.

### 4.6. Multi-Epitope Vaccine Candidate Design

The selected antigenic, promiscuous, strongly binding HTL epitopes, high-scoring LBL epitopes, and high-scoring CTL epitopes (with combined scores above the 1.0 threshold) were combined using GPGPG, KK, and AAY linkers, respectively. The EAAAK linker sequence was used to incorporate the *Mycobacterium tuberculosis* 50S ribosomal protein L7/L12 (P9WHE3), a TLR4 agonist, at the amino-terminal of the designed chimeric vaccine candidate to act as a built-in adjuvant [31]. This linker has been reported to improve protein expression, stability, and biological activity [59]. The GPGPG linker has been used to minimize junctional epitope creation in epitope-based vaccines—enhancing the processing and presentation of antigens [57]. The pan HLA-DR epitope (PADRE) and cell-penetrating TAT peptide sequences were also embedded between the built-in adjuvant and the predicted epitopes using GGGS linkers. The GGGS linker enhances the flexibility and ensures the operational separation of different epitopes [82]. Similarly, for the linear B-epitopes, KK linkers, which have been reported to maintain the independent immune reactivity of each epitope, were used [83]. In summary, all of the incorporated linkers played pivotal roles in providing an extended conformation (flexibility), assisting folding, separating protein domains, and generally making the recombinant multi-epitope vaccine structure more stable [84]. Finally, a 10xHis-tag was added at the C-terminal of the multi-epitope vaccine candidate for application in identification and purification procedures after the HRV 3C protease cleavage sequence.

### 4.7. Physicochemical Properties and Solubility Prediction

The ProtParam web server (https://web.expasy.org/protparam/) (accessed on 12 September 2024) was used to predict the physicochemical properties of the designed multi-epitope chimeric vaccine candidate. The server generates information on the protein molecular weight, aliphatic index, theoretical pI, and predicted half-lives upon expression in *Escherichia coli*, yeast, and mammal cells. In addition, information on the instability index, amino acid, and atomic composition is predicted [85]. Since *E. coli* is a common first choice for protein expression [86], the chimera’s solubility upon expression in bacteria was also predicted using the NetSolP-1.0 server (https://services.healthtech.dtu.dk/services/NetSolP-1.0/) (accessed on 12 September 2024). The NetSolP-1.0 server is used to predict the solubility and usability of purification of proteins expressed in *Escherichia coli* directly from the sequence. The working algorithm deployed in NetSolP is based on deep learning protein language models, and the server achieves state-of-the-art performance and improves the extrapolation across datasets [87]. In addition, DeepSoluE (http://lab.malab.cn/~wangchao/softs/DeepSoluE/) (accessed on 12 September 2024), which performs protein solubility prediction using a long short-term memory (LSTM) network with hybrid features comprising physicochemical properties and the distributed representation of each amino acid, was also used to predict the solubility of the vaccine candidate upon expression in bacteria [88].

### 4.8. Antigenicity, Immunogenicity, Allergenicity, and Toxigenicity Prediction

VaxiJen v2.0 (previously described) and the ANTIGENpro server from the Scratch Proteomics Predictor (https://scratch.proteomics.ics.uci.edu/) (accessed on 12 September 2024) were used to predict the antigenicity of the designed multi-epitope chimeric antigen. The ANTIGENpro server, like VaxiJen v2.0, is an alignment-free, sequence-based prediction tool that was trained on a sizable, non-redundant dataset obtained primarily from protein microarray data investigation. The VaxiJen v2.0 server achieved 76% estimated accuracy with the combined dataset in cross-validation experiments [89]. Additionally, the Vaxi-DL server (https://vac.kamalrawal.in/vaxidl/) (accessed on 12 September 2024) was used to further assess the potential of the designed candidate to stimulate protective immune responses. The Vaxi-DL server uses deep learning (DL) algorithms to classify given protein sequences into vaccine candidates and non-vaccine candidates [90].

The AllergenFP v1.0 (https://ddg-pharmfac.net/AllergenFP/) (accessed on 12 September 2024) and AllerTop v.2.0 (https://www.ddg-pharmfac.net/AllerTOP/) (accessed on 11 September 2024) servers were used to predict the potential of the designed multi-epitope antigen to stimulate allergenic responses. The AllergenFP server uses an alignment-independent descriptor-based fingerprint approach to classify proteins based on their allergenicity by implementing a four-step process. The AllergenFP server achieved accuracy of 88% with a Matthews correlation coefficient of 0.759 when analyzed using a dataset of 2427 known non-allergens and 2427 allergens [91]. The AllerTop v2.0 server operates using amino acid E-descriptors, auto- and cross-covariance transformation, and several machine learning methods, with k-nearest neighbors (kNN) performing the best (85.3% accuracy at 5-fold cross-validation) [92].

Meanwhile, the ToxinPred3 (https://webs.iiitd.edu.in/raghava/toxinpred3/) (accessed on 11 September 2024) and ToxDL (http://www.csbio.sjtu.edu.cn/bioinf/ToxDL/) (accessed on 11 September 2024) servers were used to predict the potential for toxic peptide presence in the designed chimeric antigen. The ToxinPred3 server is based on machine learning (ML) and deep learning methods and has been trained using a large dataset of 5518 toxic and 5518 non-toxic experimentally validated peptides [93]. The ToxDL server uses a deep learning-based alignment-free approach for the in silico prediction of protein toxicity, and independent test results showed that ToxDL outperformed servers based on traditional homology-based approaches and state-of-the-art machine learning techniques [94].

### 4.9. Secondary Structure and Intrinsic Disorder Prediction

The PSIPRED v4.0 server (http://bioinf.cs.ucl.ac.uk/psipred) (accessed on 11 September 2024) was used to predict the secondary structure of the designed chimeric antigen. The PSIPRED algorithm incorporates two feed-forward neural networks that analyze the output of a Position-Specific Iterated BLAST (PSI-BLAST) to accurately predict a protein’s secondary structure [95]. The PSIPRED v4.0 server has Q3 prediction accuracy of 84.2.5% [96].

Disordered proteins constitute an important class of antigens in a wide range of human pathogens, and they are frequently the targets of protective immune responses [97]. Intrinsically disordered regions (IDRs) in the designed chimeric vaccine candidate were predicted using the IUPred3 (https://iupred.elte.hu/) (accessed on 11 September 2024) and PrDOS (https://prdos.hgc.jp/cgi-bin/top.cgi) (accessed on 12 September 2024) servers. The IUPred web server algorithm employs the amino acid sequence or UniProt ID to predict the tendency for each amino acid to be in a disordered region using a unique energy estimation approach [98]. The PrDOS server algorithm is composed of two predictors (one based on the local amino acid sequence and the other based on template proteins) to predict protein disorder using support vector machine (SVM) and PSI-BLAST [99].

### 4.10. Antibody Class Prediction

Certain immunoglobulins, principally IgG, have been implicated in protective immune responses against *Mycobacterium tuberculosis* [100]. The Antibody Class(es) Predictor for Epitopes (AbCPE) server (http://bioinfo.unipune.ac.in/AbCPE/Home.html) (accessed on 12 September 2024) was used to investigate the antibody classes elicited by the selected linear B-epitope sequences incorporated into the designed vaccine candidate. The AbCPE server is an innovative method based on a multi-label classification algorithm (including binary relevance, label powerset, random forest, and AdaBoost) for antibody class(es) prediction and the binding of peptide sequences to IgG, IgE, IgA, and IgM [101].

### 4.11. IFN-γ-Inducing Epitope Prediction

Interferon-gamma (IFN-γ), which is vital for the generation of protective responses against intracellular pathogens like *M. tuberculosis*, is reported to be important in the development of protective anti-TB immune responses [102]. IFN-γ epitopes present in the designed chimeric antigen were predicted using the hybrid motif and SVM approach on the IFNepitope server (https://webs.iiitd.edu.in/raghava/ifnepitope/scan.php) (accessed on 12 September 2024). The IFNepitope server algorithm functions by constructing overlapping 15-mer sequences, which are used to predict the IFN-γ epitopes. The server was developed using a training dataset of experimentally validated MHC class II binders obtained from the Immune Epitope Database (IEDB), comprising 3705 IFN-γ inducing and 6728 non-inducing MHC class II binders [103].

### 4.12. Prediction of Epitopes for Mouse MHC II Alleles

To examine the potential of the designed vaccine candidate to stimulate immune responses in preclinical studies, 15-mer MHC II binding peptides for mouse alleles in the designed vaccine candidate were predicted using the NetMHCII 2.3 server (https://services.healthtech.dtu.dk/service.php?NetMHCII-2.3) (accessed on 12 September 2024). This server uses artificial neural networks to predict the binding affinities to molecules covering human alleles (HLA-DR, HLA-DQ, and HLA-DP) as well as 7 mouse alleles. The operation of this server has been described above.

### 4.13. Immune Simulation Analyses

The immune response elicited by the chimeric vaccine candidate was predicted by the C-ImmSim server (https://kraken.iac.rm.cnr.it/C-IMMSIM/index.php) (accessed on 12 September 2024), which performs in silico immune simulation analyses. The C-ImmSim server is an innovative agent-based model that employs a position-specific scoring matrix (PSSM) for epitope prediction and machine-learning approaches to predict immunological interactions. The server “simultaneously simulates three compartments that represent three separate anatomical regions found in mammals: (i) the bone marrow, where hematopoietic stem cells are simulated and produce new lymphoid and myeloid cells; (ii) the thymus, where naive T cells are selected to avoid autoimmunity; and (iii) a tertiary lymphatic organ, such as a lymph node” [104]. A total of 500 simulation steps were set. The default server simulation parameters were used, with the three time steps set at 1, 84, and 168 for the three injections (each time step is 8 h, and time step 1 is injection at time = 0). These settings were based on a slight modification of the protocol used in phase II clinical trials for the M72/AS01E vaccine candidate [105].

### 4.14. Tertiary Structure Prediction, Refinement, and Validation

The 3D structure of the designed chimeric antigen was predicted using ColabFold, an easy-to-use interface that uses the AlphaFold2 [106] technology within the Google Colab environment. ColabFold offers the accelerated prediction of protein structures and complexes by combining the fast homology search of MMseqs2 v17.0 with AlphaFold2 or RoseTTAFold2 [107]. The 3D structure obtained was refined using the GalaxyRefine server (http://galaxy.seoklab.org/cgi-bin/submit.cgi?type=REFINE) (accessed on 12 September 2024). The GalaxyRefine server functions based on a refining technique that was successfully evaluated in CASP10 community-wide tests and found to greatly improve the local structure quality compared to other servers. To accomplish overall structure relaxation, this technique first rebuilds side chains, performs side chain repacking, and then employs a molecular dynamics simulation [108]. The overall quality score of the refined chimera 3D structure was assessed using the ProSA-web (https://prosa.services.came.sbg.ac.at/prosa.php, accessed on 12 September 2024) and ERRAT (http://services.mbi.ucla.edu/ERRAT/, accessed on 12 September 2024) servers. ProSA is widely used to examine 3D models of protein structures for potential errors and is used for error recognition in experimentally determined structures, theoretical models, and protein engineering [109]. The ERRAT server, on the other hand, analyzes non-bonded atom–atom interactions compared to reliable high-resolution crystallography structures [110].

### 4.15. Discontinuous B-Cell Epitope Prediction

Following the validation of the 3D structure of the modeled chimeric vaccine candidate, the ElliPro (http://tools.iedb.org/ellipro/) server (accessed on 13 September 2024) was deployed to predict discontinuous B-cell epitopes using the default parameters. The ElliPro server “implements a modified version of Thornton’s method and, together with a residue clustering algorithm, the MODELLER v10.6 program and the Jmol viewer v14.0, allows the prediction and visualization of antibody epitopes in protein sequences and structures”. The ElliPro server has been tested on a benchmark dataset of conformational epitopes deduced from the 3D structures of antibody–protein complexes. ElliPro produced the best performance when compared with other structure-based epitope prediction algorithms, giving an AUC value of 0.732 when considering the most significant prediction for each protein [111].

### 4.16. Binding Pocket Prediction, Molecular Docking, Binding Affinity, and Interaction Analyses

To assess the binding pockets in the designed chimera, the CASTp 3.0 (http://sts.bioe.uic.edu/castp/index.html?2was) server (accessed on 13 September 2024) was used. CASTp is a web server that uses the alpha shape method developed in computational geometry to locate, delineate, and measure different geometric and topological properties of protein structures [112].

To assess the interaction between the designed chimera and TLR4, a protein–protein docking analysis was executed on the ClusPro 2.0 webserver (https://cluspro.bu.edu/login.php) (accessed on 13 September 2024). The ClusPro 2.0 server has constantly been among the best docking servers, as established in the Critical Assessment of Predicted Interactions (CAPRI), delivering high predictive performance for the docking of protein–protein complexes [113]. The server performs docking through the following computational steps: (1) rigid body docking by sampling billions of conformations, (2) root mean square deviation (RMSD)-based clustering of the 1000 lowest-energy structures generated to find the largest clusters that will represent the most likely models of the complex, and lastly (3) the refinement of selected structures through energy minimization. The rigid body docking step uses PIPER, a docking program based on the Fast Fourier Transform (FFT) correlation approach. The server performs similarity-based docking by retrieving templates from a database of experimentally determined structures and building models using energy-based optimization, which allows for structural flexibility [113].

The binding affinity of the predicted interaction between the vaccine candidate and the TLR4 immune receptor was assessed using the Protein Binding Energy (PRODIGY) prediction server (https://wenmr.science.uu.nl/prodigy/) (accessed on 13 September 2024). This server implements a simple but very efficient predictive model based on intermolecular contacts and properties derived from the non-interface surface [33]. The docking results were further analyzed by the PDBsum server (http://www.ebi.ac.uk/thornton-srv/databases/pdbsum/Generate.html) (accessed on 13 September 2024), which generates a 2D picture of the interactions at the ligand–protein interface from the 3D coordinates [114].

### 4.17. Normal-Mode Analyses

The iMODS server (http://imods.Chaconlab.org/) (accessed on 13 September 2024) was used to perform normal-mode analysis (NMA) for the docked vaccine candidate–TLR4 complex. This analysis was performed to establish the internal dihedral coordinates while measuring the cooperative functional movements in the docked complex. The fundamental dynamic simulation program of the iMODS server was used to guarantee the energy minimization, molecular stability, and atomic mobility of the designed vaccine candidate in the docked complex. The iMODS server determined the possible motions of the vaccine candidate–TLR4 complex using specialized parameters, including B-factors, the RMSD, complex deformability, eigenvalues, and covariance values, as well as elastic models [115].

### 4.18. Codon Optimization and In Silico Cloning

The K12 *E. coli* strain was selected to express the multi-epitope chimeric antigen, since bacteria are often the primary host for protein expression experiments. The Java Codon Adaptation Tool (https://www.jcat.de/) (accessed on 14 September 2024) was used to reverse-translate the protein sequence into a DNA sequence and perform codon optimization. The JCat server offers the flexibility to adapt the codon usage for any gene that the user provides to that of any host organism. The server also provides information on the codon adaptation index (CAI) and GC% content, which have both been reported to affect protein expression in bacteria [116,117]. For the expression of the designed antigen in *E. coli*, the codon-optimized gene sequence was inserted between the pET30a+ vector NdeI and XhoI restriction sites using the Snapgene software (https://www.snapgene.com/) (accessed on 14 September 2024).

## Figures and Tables

**Figure 1 ijms-26-01587-f001:**
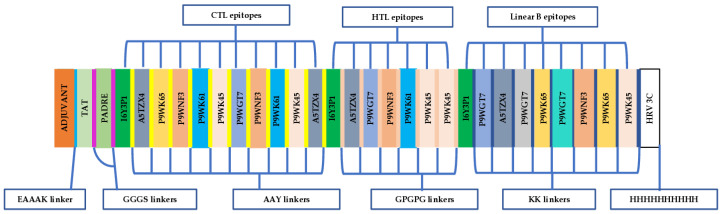
Graphic representation of the designed multi-epitope chimeric vaccine candidate, TB-MEVA−1. The 683-amino-acid polypeptide sequence containing a built-in adjuvant (orange) at the N-terminal linked to the multi-epitope sequence through an EAAAK linker (light blue) to the PADRE and TAT peptides, which are joined through GGGS linkers. Antigenic high-affinity HTL epitopes, as well as high-scoring LBL and CTL epitopes, are joined by AAY (yellow), GPGPG (light orange), and KK linkers (blue), respectively). A 10×His tag is added at the C-terminal to facilitate purification and identification, separated from the peptide epitopes using the HRV 3C protease cleavage sequence.

**Figure 2 ijms-26-01587-f002:**
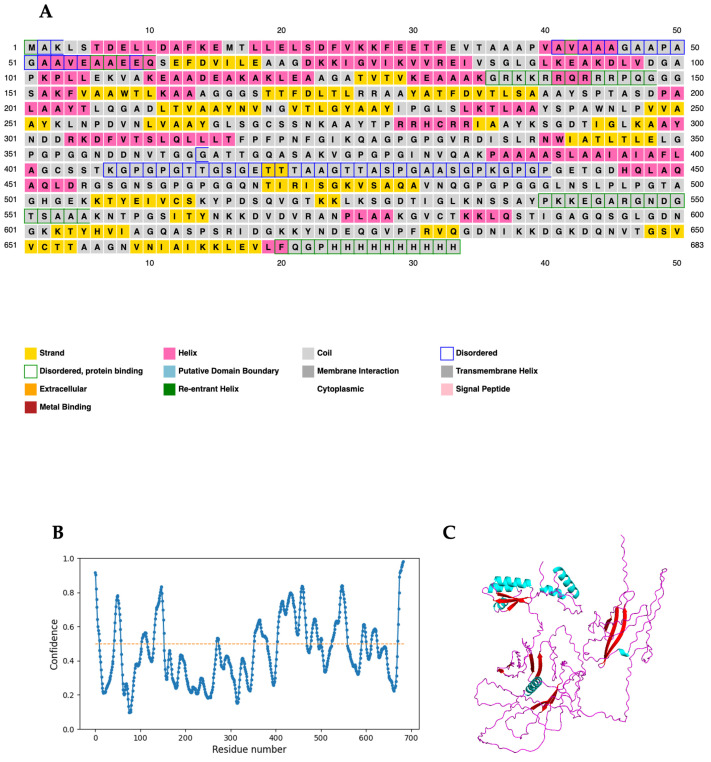
Graphic representation of key secondary structure types of the designed subunit vaccine sequence. (**A**) The protein is predicted to comprise alpha helices (27.0%), beta strands (13.0%), and coils (58.0%), and (**B**) 31% of the amino acid residues are predicted by the DISOPRED3 server to be located in intrinsically unstructured regions. The dashed orange line indicates the disorder threshold (**C**). The 3D structure of the designed vaccine candidate shows the different secondary structure features, including helices (cyan), coils (red), and strands (magenta).

**Figure 3 ijms-26-01587-f003:**
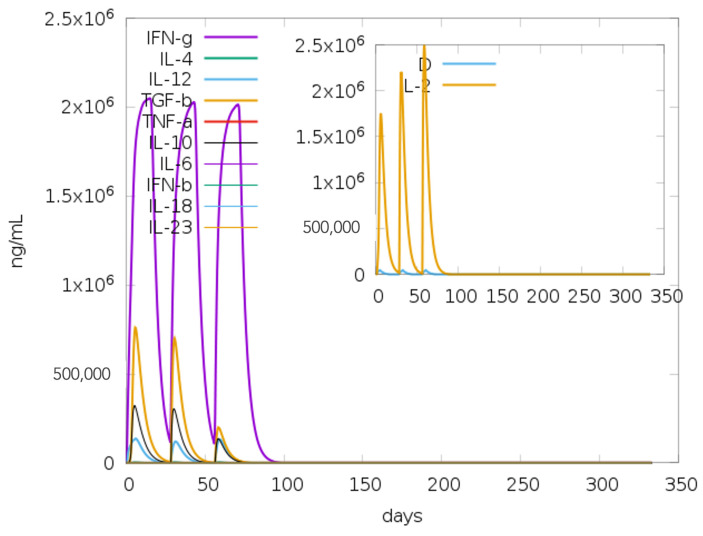
Immunological simulation of the cytokine profile following three TB-MEVA−1 injections spaced 28 days apart. Following the three doses, the primary plot displays varying cytokine levels. The blue line in the insert figure represents the Simpson index, D, and the IL-2 level. D is a diversity metric. An increase in D over time denotes the emergence of distinct epitope-specific dominant T-cell clones. The diversity decreases with the D value.

**Figure 4 ijms-26-01587-f004:**
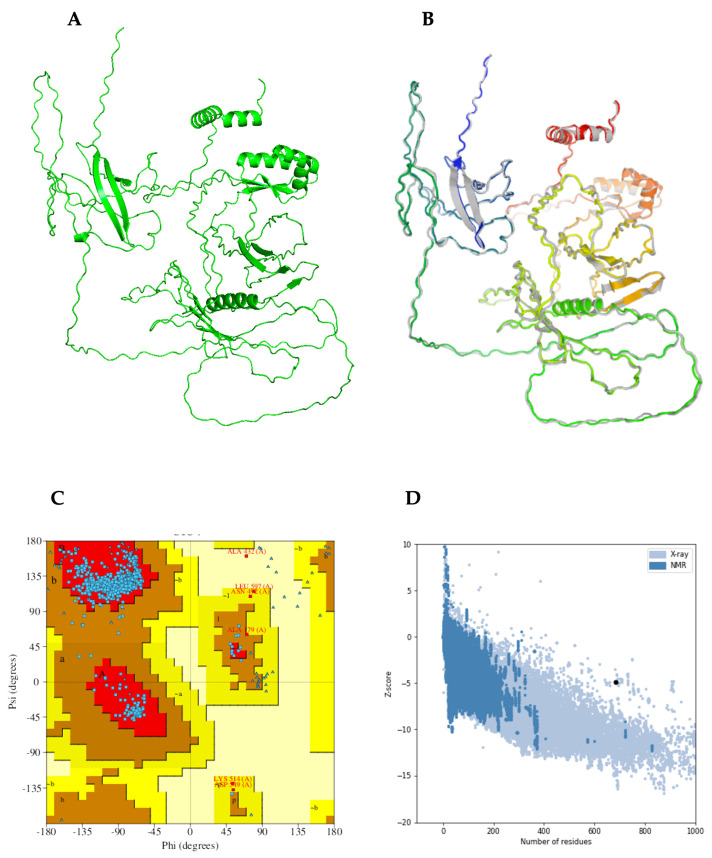
Protein modeling, refinement, and validation. (**A**) The initial 3D model of the multi-epitope vaccine candidate predicted on the ColabFold server. (**B**) Superimposition of the refined 3D structure (colored) on the initial “crude model” (gray) by the GalaxyRefine server. Validation of the refined model showing (**C**) Ramachandran plot analysis with 97.1% of residues in favored regions (A, B, L), 2.1% in allowed regions (a, b, l, p and ~a, ~b, ~l, ~p), and 0.2% of protein residues in disallowed (outlier) regions, and (**D**) the ProSA-web plot, with a Z-score of −4.92.

**Figure 5 ijms-26-01587-f005:**
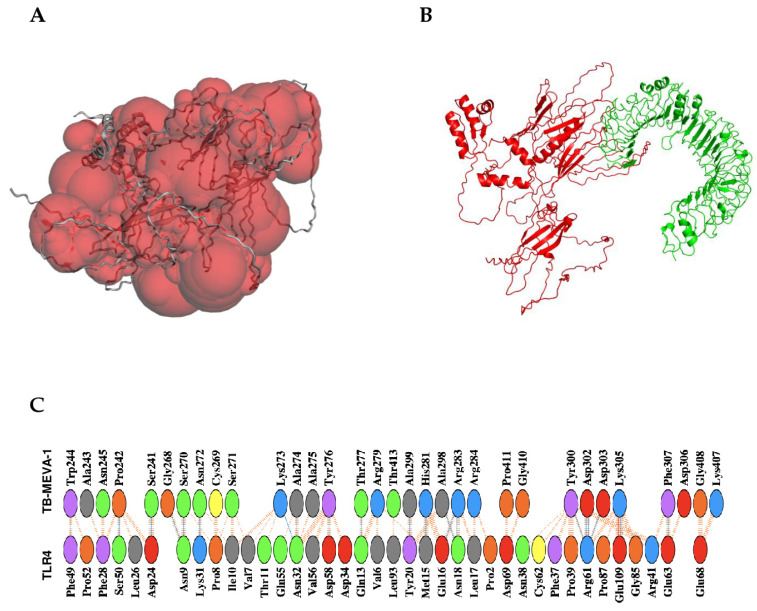
Protein–protein interaction pocket prediction and molecular docking of vaccine candidate with the TLR4 receptor. (**A**) The largest predicted PPI pocket located at the interface of the designed chimeric vaccine candidate (red), with a surface area of 29,113.275 Å^2^ and a surface volume of 129,803.080 Å^3^. (**B**) The docked complex of the designed vaccine candidate (red) with the Toll-like receptor 4 chain (green). (**C**) Interaction network of the vaccine candidate and TLR4, respectively; hydrogen bonding interactions are shown in blue. Non-bonded interactions and salt bridges are shown in orange. The positive, negative, neutral, aliphatic, and aromatic charged residues are shown in blue, red, green, gray, and purple colors, respectively.

**Figure 6 ijms-26-01587-f006:**
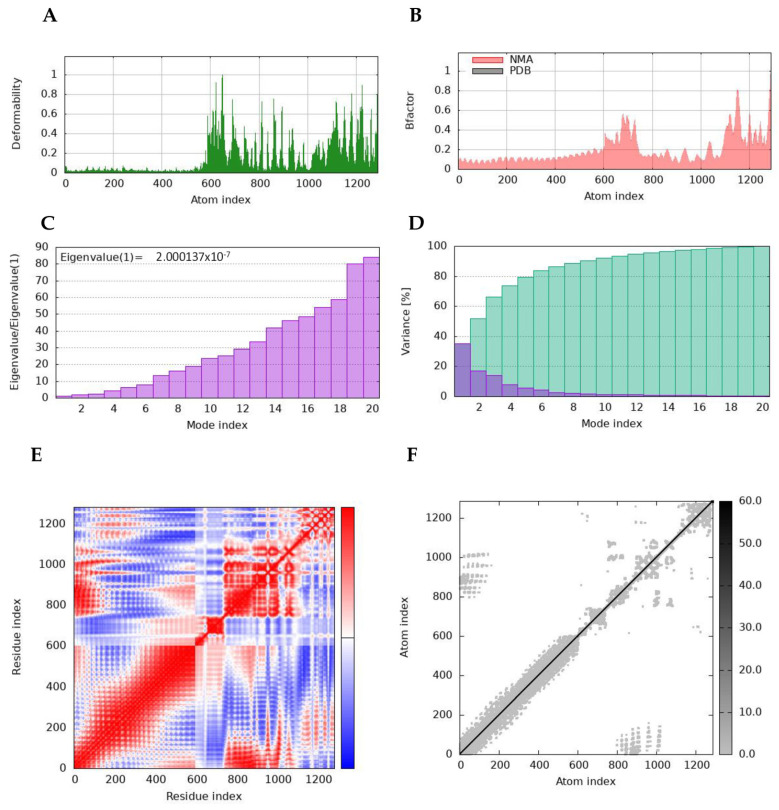
Molecular dynamics simulation analysis of the docked vaccine candidate–TLR4 complex. (**A**) The main-chain deformability of the TB-MEVA−1-TLR4 complex. (**B**) The B-factor quantifies the uncertainty of each atom. (**C**) The eigenvalue stipulates the movement stiffness linked to the normal modes. (**D**) The variance map is related to the individual variances (red) and the cumulative variances (green). (**E**) The covariance graph between pairs of residues shows the correlated (red), uncorrelated (white), or anti-correlated (blue) mobility of the pairs. (**F**) The elastic network model portrays the atom pairs linked by springs, with the darker grays indicating the spring stiffness.

**Figure 7 ijms-26-01587-f007:**
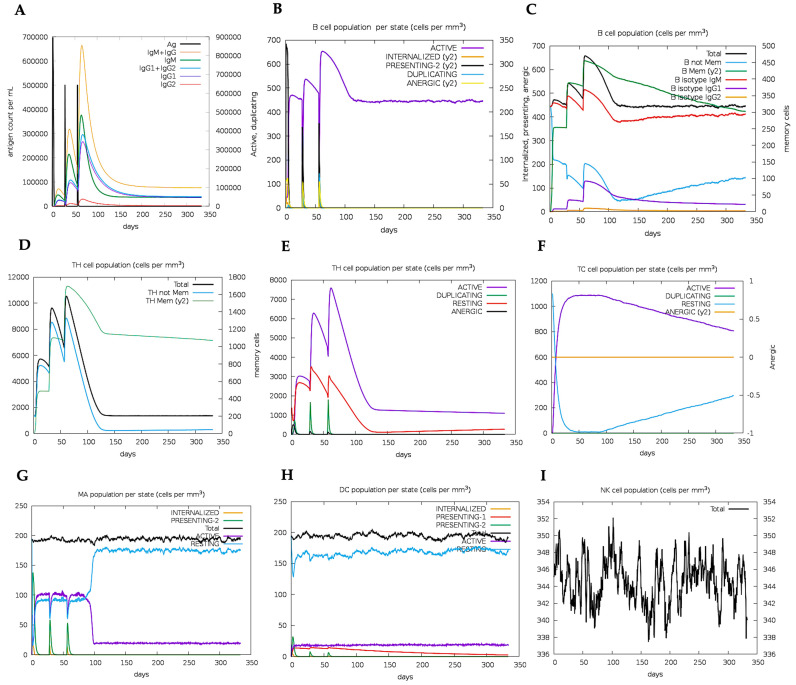
In silico model of elicited immune response by TB-MEVA−1, showing the (**A**) antigen and induced immunoglobulins, (**B**) B-cell population/state, (**C**) B-cell population, (**D**) TH cell population, (**E**) TH cell population/state, (**F**) TC cell population/state, (**G**) MA population per state, (**H**) dendritic cell population/state, and (**I**) natural killer cell population.

**Figure 8 ijms-26-01587-f008:**
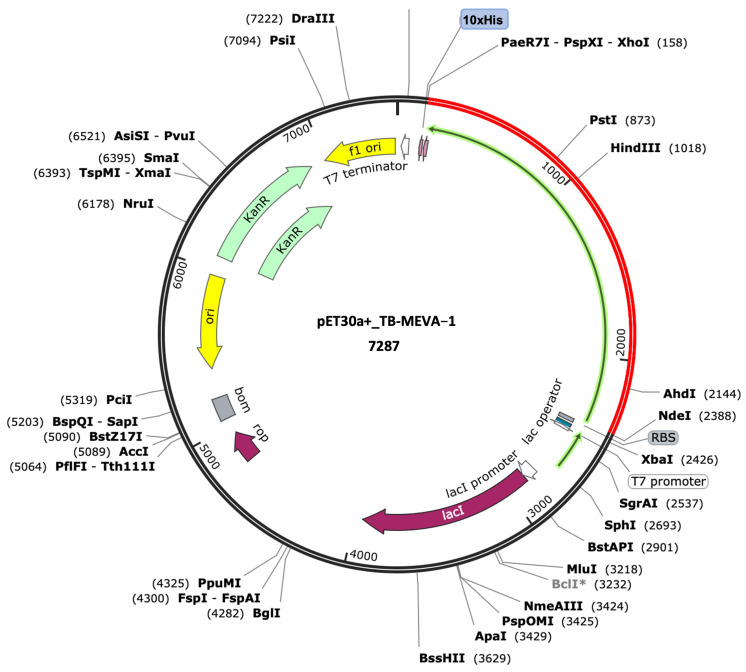
Computational cloning of the gene coding for the designed vaccine candidate into the pET30a (+) expression vector. The red section denotes the gene coding for the vaccine candidate, while the black circle denotes the pET28(a)+ backbone. The 10xHis tag is placed at the carboxy-terminus of the vaccine candidate. The * in the BclI restriction enzyme indicates that the enzyme displays star activity.

**Figure 9 ijms-26-01587-f009:**
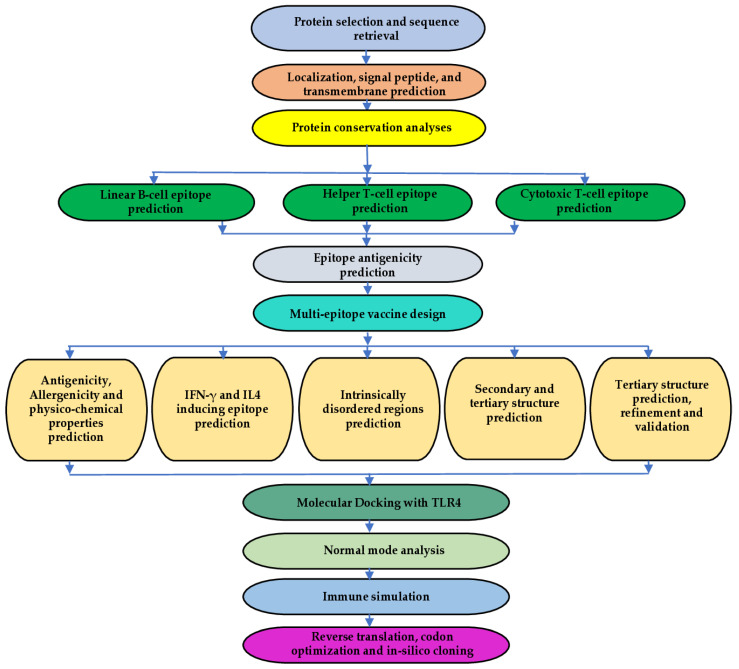
Workflow for the study. The approach began with identifying the target lipoproteins and carrying out a conservation analysis. CTL, HTL, and LBL epitope prediction; vaccine design, 3D structure modeling, refinement, and verification followed. Molecular docking of the validated 3D model with TLR4, molecular dynamics simulation, and in silico immune simulation studies was performed to investigate the potential of the vaccine candidate to elicit immune responses. Finally, reverse translation, codon optimization, and in silico cloning were performed.

**Table 1 ijms-26-01587-t001:** Protein localization prediction.

Protein	Localization	
	DeepLocPro-1.0	TBpred
P9WK45	Cytoplasmic membrane	Protein attached to membrane by lipid anchor
P9WNF3	Cytoplasmic membrane	Secreted protein
P9WK65	Cytoplasmic membrane and cell surface	Protein attached to membrane by lipid anchor
A5TZX4	Cytoplasmic membrane	Integral membrane protein
I6Y3P1	Cytoplasmic membrane	Cytoplasmic protein
P9WGT7	Cytoplasmic membrane	Protein attached to membrane by lipid anchor
P9WK61	Cytoplasmic membrane	Protein attached to membrane by lipid anchor

**Table 2 ijms-26-01587-t002:** Conservation of target proteins in selected *Mycobacterium* species.

Protein	Percentage Identity (%)							
	*M. decipiens*	*M. leprae*	*M. lacus*	*M. gordonae*	*M. asiaticum*	*M. bovis*	*M. riyadhense*	*M. pseudokansasii*
P9WK45	91.5	68.1	81.0	70.6	72.5	100	77.5	81.8
P9WNF3	86.0	Not found	83.3	77.2	75.0	100	65.8	75.0
P9WK65	71.6	76.4	75.9	66.4	30.9	100	67.3	62.8
A5TZX4	84.3	61.2	37.7	63.5	60.5	99.8	62.0	60.0
I6Y3P1	83.1	Not found	85.6	79.9	79.7	100	85.3	79.1
P9WGT7	91.1	77.6	85.1	81.5	80.5	100	84.9	82.5
P9WK61	87.6	43.0	81.1	80.5	69.8	100	85.7	82.5

**Table 3 ijms-26-01587-t003:** Predicted epitopes incorporated into the designed multi-epitope vaccine candidate.

Antigen	LBL	CTL	HTL
I6Y3P1	GGLNSLPLPGTAGHGE	TTFDLTLRR	NDDRKDFVTSLQLLLTFPFPNFGIKQA
A5TZX4	LKSGDTIGLKNSSAYP	ATFDVTLSA WIATLTLEL KSGDTIGLK	VRDISLRNWIATLTLEL
P9WGT7	TYEIVCSKYPDSQVGT EGARGNDGTSAAAKNTPGSITYN LQSTIGAGQSGLGDNG	ATTGQASAK NLDGPTLAK SPAWNLPVV	GNDDNVTGGGATTGQASAKV
P9WK65	DVDVRANPLAAKGVCT YNDEQGVPFRVQGDNI	SPTASDPAL	
P9WNF3	TYHVIAGQASPSRIDG	TLQGADLTV KLNPDVNLV FLAGCSSTK TLTSALSGK	INVQAKPAAAASLAAIAIAFLAGCSSTK
P9WK61	GSVVCTTAAGNVNIAI	NVNGVTLGY GLSGCSSNK	DGKDQNVTGSVVCTTAAGNV TTGSGETTTAAGTTASPGAASGPK
P9WK45		IPGLSLKTL TPRRHCRRI	ETGDHQLAQAQLDRGSGNS GQNTIRISGKVSAQAVNQ

Overlapping sequences are underlined.

**Table 4 ijms-26-01587-t004:** Immunoglobulin (Ig) class induction by linear B-lymphocyte epitopes.

Predicted LBL Epitope	Predicted Epitope Probability (%)	Predicted Antibody Class			
		IgG	IgE	IgA	IgM
GGLNSLPLPGTAGHGE	66	+	−	−	−
TYEIVCSKYPDSQVGT	66	+	−	−	−
LKSGDTIGLKNSSAYP	100	+	−	+	−
EGARGNDGTSAAAKNTPGSITYN	100	+	−	−	−
DVDVRANPLAAKGVCT	66	+	−	+	−
LQSTIGAGQSGLGDNG	100	+	−	−	−
TYHVIAGQASPSRIDG	66	+	−	−	−
YNDEQGVPFRVQGDNI	100	+	−	−	−
GSVVCTTAAGNVNIAI	66	+	−	−	−

+: Epitope/peptide binds to the respective antibody class. −: Epitope/peptide does not bind to the respective antibody class.

**Table 5 ijms-26-01587-t005:** The physicochemical properties of the designed chimeric vaccine.

Property	Measurement
Number of Amino Acids	683
Molecular Weight	69.7kDa
Formula	C_3060_H_4917_N_885_O_959_S_8_
Theoretical pI	9.36
Instability Index	27.03
Aliphatic Index (AI)	−0.295
Grand Average of Hydropathicity (GRAVY)	76.31
Solubility on Expression (DeepSoluE)	0.4218
Solubility on Expression (NetSolP 1.0)	0.4811

**Table 6 ijms-26-01587-t006:** The antigenicity, allergenicity, and toxigenicity of the designed vaccine candidate.

Property	Measurement	Remark
Antigenicity (VaxiJen v2.0)	1.1314	Antigenic
Antigenicity (ANTIGENpro)	0.9218	Antigenic
Antigenicity (VaxiDL)	96.66%	Antigenic
Allergenicity (AllerTOP v2.0)	Probable non-allergen	Probable non-allergen
Allergenicity (AllergenFP 2.0)	Probable non-allergen	Probable non-allergen
Toxicity (ToxinPred)	Non-toxin	Non-toxigenic
Toxicity (ToxDL)	0.0001	Non-toxigenic

**Table 7 ijms-26-01587-t007:** Number of predicted peptides binding to mouse H2 class II alleles.

Allele	H-2-IAb	H-2-IAd	H-2-IAk	H-2-IAs	H-2-IAu	H-2-IEd	H-2-IEk
Number of epitopes	78	32	9	22	24	3	0

## Data Availability

Data are contained within the article’s figures, tables, and Appendix A.

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
