# Peer review of "One Health Approach to the Computational Design of a Lipoprotein-Based Multi-Epitope Vaccine Against Human and Livestock Tuberculosis"

_ijms, 2025, doi:10.3390/ijms26041587_

Round 1

Reviewer 1 Report

Comments and Suggestions for Authors

This is very promising research, as it suggests that it may be possible to create a new vaccine against tuberculosis (TB), which has remained a major health hazard and for which there has been no effective method of immunisation other than BCG, at a very low cost and in a very simple way. Given the global spread of TB, such low-tech methods are desirable. However, it goes without saying that through both in-vitro and in-vivo investigations are needed, and I hope this will happen soon. I have minimal comments, which I hope will be quickly rectified and the next step taken. Anyway, I have no doubt that this is a promising approach.

Names of L113 lipoproteins: they seem to be named differently from [28]. It would be easier to understand if the auhthors  could unify them. Also, please explain w hy lipoproteins were targeted. Perhaps because they are easily presented on the outer wall of the bacteria, also, the toxicity is probably because it is easy to get out of the bacteria, and so on.

Of course, the designed chimeric antigen extracted would not function as a lipoprotein,

However, it is possible that it may be highly cytotoxic (although this is denied by Table 6).

L167 The sudden appearance of CTLs in the table should be explained in L154.

Fig 1 It is difficult to understand. For example, epitope should be all the way up and linker should be all the way down.

There is no point in mixing up the top and bottom.

I think there is a reason why there are three different types of linker, which should be described.

Also, is it necessary to make it so colourful? It should also take colourblind readers into account.

Fig. 2 C shows that it does not seem to have the transmembrane domain that some of the original lipoporoteins had (?).

Does one coil and coil make it?

Did they dare to avoid this? It would be nice if you could write something about the situation.

The different colours waved for the structure in A and C make it difficult to see, in A the coil is grey but in C it is red. This is difficult to see and should be improved.

Author Response

Comment: This is very promising research, as it suggests that it may be possible to create a new vaccine against tuberculosis (TB), which has remained a major health hazard and for which there has been no effective method of immunisation other than BCG, at a very low cost and in a very simple way. Given the global spread of TB, such low-tech methods are desirable. However, it goes without saying that through both in-vitro and in-vivo investigations are needed, and I hope this will happen soon. I have minimal comments, which I hope will be quickly rectified and the next step taken. Anyway, I have no doubt that this is a promising approach.

Response: We appreciate the reviewer's comments and agree that there is a need to validate the results from this study using in-vivo and in-vitro models. We want to confirm that this is just the beginning of a bigger project, and we have the wet lab studies planned.

Comment: Names of L113 lipoproteins: they seem to be named differently from [28]. It would be easier to understand if the authors could unify them. Also, please explain why lipoproteins were targeted. Perhaps because they are easily presented on the outer wall of the bacteria. Also, the toxicity is probably because it is easy to get out of the bacteria, and so on.

Response: We greatly appreciate this comment from the reviewer. We agree with the reviewer that the lipoprotein names are different from what is indicated in the cited article. However, we would like to indicate that since the selected antigens were retrieved from two different sources, we tried to use a method of identification that is very harmonized – in this case, the accession numbers, each of which specifically identifies each vaccine candidate.

Concerning the choice of the vaccine candidates, lipoproteins, as bacterial membrane-anchored proteins, have a variety of biological functions. These functions include acting as bacterial virulence factors, recognizing and activating the host’s immune system, and other diverse roles. They have, thus, been among the most popular drug targets and vaccine candidates1. Information on the rationale for the selection of lipoproteins is found in L105-L109 of the manuscript. We also agree with the reviewer that their localization on the membrane surface is also an important factor.

Concerning toxicity, our prediction analyses indicated the designed chimera is not toxic. Nevertheless, there is a need for in-vitro and in-vivo validation of these results like other results obtained in this work since this first part of the study is primarily in-silico.

Comment: Of course, the designed chimeric antigen extracted would not function as a lipoprotein,

However, it is possible that it may be highly cytotoxic (although this is denied by Table 6).

Response: Concerning possible toxicity, our toxigenicity prediction on two different servers suggests that the designed chimera is not toxic. Nevertheless, there is a need for in-vitro and in-vivo validation of these results, like other results obtained in this work, since this first part of the study is primarily in-silico.

Comment: L167 The sudden appearance of CTLs in the table should be explained in L154.

Response: We appreciate this comment and have implemented the correction. CTL which stands for cytotoxic T-lymphocyte has been added to the subtopic in L151

Comment: Fig 1 It is difficult to understand. For example, epitope should be all the way up and linker should be all the way down.

There is no point in mixing up the top and bottom.

I think there is a reason why there are three different types of linker, which should be described.

Also, is it necessary to make it so colourful? It should also take colourblind readers into account.

Response: We appreciate these comments from the reviewer. Figure 1 has been adjusted following the recommendations from the reviewer. The linkers have been placed at the bottom, while the epitopes have been placed at the top. In addition, the colour scheme of the figure has been adjusted to make it more colour-friendly. Adjustments have been made to ensure that epitopes from the same antigens have the same colour.

Concerning the linkers, we agree with the reviewer that different linkers have been used for a reason. A description of the reason for using the different linker is found in L626-L641 as follows:

The EAAAK linker sequence was used to incorporate the Mycobacterium tuberculosis 50S ribosomal protein L7/L12 (P9WHE3), a TLR4 agonist at the amino-terminal of the designed chimeric vaccine candidate to act as a built-in adjuvant 2. This linker has been reported to improve protein expression, stability, and biological activity 3. The GPGPG linker has been used to minimize junctional epitope creation in epitope-based vaccines – enhancing the processing and presentation of antigens 4. The pan HLA-DR epitope (PADRE) and cell-penetrating TAT peptide sequences were also embedded between the built-in adjuvant and the predicted epitopes using GGGS linkers. The GGGS linker enhances flexibility and ensures the operational separation of different epitopes 5. Similarly, for the linear B-epitopes, KK linkers, which have been reported to maintain the independent immune reactivity of each epitope, were used 6. In summary, all the incorporated linkers play pivotal roles in providing an extended conformation (flexibility), assisting folding, separating protein domains, and generally making the recombinant multi-epitope vaccine structure more stable 7.

Comment: Fig. 2 C shows that it does not seem to have the transmembrane domain that some of the original lipoporoteins had (?).

Does one coil and coil make it?

Did they dare to avoid this? It would be nice if you could write something about the situation.

The different colours waved for the structure in A and C make it difficult to see, in A the coil is grey but in C it is red. This is difficult to see and should be improved.

Response: We appreciate these comments from the reviewer. The designed chimera does not have any transmembrane domains. This is because only epitopes from the selected antigens were incorporated in the chimeric vaccine candidate. Concerning the secondary structure features, we would like to indicate the server used predicted most of the residues to be found in coiled regions. However, the protein was also predicted to contain helices and strands. This information is found in L227-L228 of the revised manuscript.

Concerning the disparity in colour between Fig. 2A and Fig. 2C, we will agree with the reviewer that there is no match in the colours. However, we would like to indicate the colour schemes were automatically generated by the different servers and tools used, and we, as authors, are unable to match the colours on the two figures. We will, therefore, seek the understanding of the reviewer that this is beyond our control.

Reviewer 2 Report

Comments and Suggestions for Authors

The computational analyses provide promising foundations, the manuscript could be improved if a more thorough discussion of the limitations of these models were conducted.  For example, some predictions, such as those on solubility or instability indices, may not fully reflect the protein's behaviour in biological systems. The manuscript provides a detailed explanation of the design and theoretical predictions; preliminary experimental results with the vaccine candidate are not discussed. A mention of initial tests in vitro or in animal models would also improve the robustness of the manuscript. Although the paper refers to the limitations of BCG and describes the development of new TB vaccines, it would be useful to include a more detailed comparison of TB-MEVA-1 with other vaccine candidates in development. This would help to provide a broader view of advantages and disadvantages.The manuscript is rather dense and could benefit from a clearer structure. For example, the transition between topics such as immune response mechanisms, structural features and computational predictions could be smoother. The use of subtitles and summaries of key results in each section would improve readability.

Comments on the Quality of English Language

A slight revision of English is recommended. 

Author Response

Comment: The computational analyses provide promising foundations, the manuscript could be improved if a more thorough discussion of the limitations of these models were conducted.  For example, some predictions, such as those on solubility or instability indices, may not fully reflect the protein's behaviour in biological systems. The manuscript provides a detailed explanation of the design and theoretical predictions; preliminary experimental results with the vaccine candidate are not discussed. A mention of initial tests in vitro or in animal models would also improve the robustness of the manuscript. Although the paper refers to the limitations of BCG and describes the development of new TB vaccines, it would be useful to include a more detailed comparison of TB-MEVA-1 with other vaccine candidates in development. This would help to provide a broader view of advantages and disadvantages. The manuscript is rather dense and could benefit from a clearer structure. For example, the transition between topics such as immune response mechanisms, structural features and computational predictions could be smoother. The use of subtitles and summaries of key results in each section would improve readability.

Response: We appreciate the comments of the reviewer and the intention to improve the quality of the manuscript. Concerning the limitations of model use, while we agree with the author that this could be valuable, we think that considering that so many servers and tools were used in this work, doing this will make the manuscript unnecessarily long. However, we have acknowledged that a major limitation of this study is the fact that all the results presented are from computational predictions and we have recognized the important need to validate the results presented here using different in-vivo and in-vitro models. This is indicated as one of the limitations of this study and have proposed the possible next steps in the manuscript (L523-L538)

Meanwhile, in line with reviewer recommendations, we had characterized and compared TB-MEVA-1 and selected vaccine candidates in terms of antigenicity, allergenicity, solubility, and instability index, but due to the voluminous nature of the article, we decided to leave this part out. We would like to indicate that since this work is purely computational, it is possible to only compare TB-MEVA-1 and vaccine candidates based on recombinant antigens.

Concerning the structure, we agree with the reviewer that transitions between topics could ease understanding. However, a big challenge we faced with this project was that several distinct analyses were performed, some of which were tricky to fit under transitions and subtopics, as suggested. In addition, given the enormous volume of the work performed, we tried as much as possible to present the key results and put some of the results in the supplementary data.